Published in FAST Workshop on Smalltalk Related Technologies (11/2022)

# Testing Finalization Mechanisms with Heap Fuzzing

**Guillermo Polito**                                                     *guillermo.polito@inria.fr*
*Univ. Lille, Inria, CNRS, Centrale Lille, UMR 9189 CRIStAL, F-59000 Lille, France*

**Reviewed on OpenReview:** *https://openreview.net/forum?id=TosPNPrDLm*

## Abstract

Producing robust memory manager implementations is a challenging task. Defects in garbage collection algorithms produce subtle effects that are revealed later in program execution as memory corruptions. This problem is exacerbated by the fact that garbage collection algorithms deal with low-level aspects to be efficient. Finding, reproducing, and debugging such bugs is complex and time-consuming.

In this article, we propose to fuzz heaps by generating large sequences of random heap events guided toward testing Pharo's ephemeron implementation. Heap events exercise the garbage collection algorithm with the objective of generating VM crashes and finding bugs. Once a bug is found, we use a test case reduction algorithm that finds the smaller subset of events that reproduces the issue.

We implemented our approach on top of the VM simulator of the Pharo VM. We used it to test the ephemeron finalization mechanism in Pharo's sequential stop-the-world generational scavenger. Our prototype found two bugs in Pharo's ephemeron implementation and one in the heap compaction algorithm. We show how such test cases were automatically reduced to trivial sequences that were easy to debug.

## 1 Introduction

Producing robust implementations of memory managers is still nowadays a challenging task. Defects in garbage collection algorithms produce subtle effects that are revealed later in program execution as memory corruptions. This problem is exacerbated by the fact that garbage collection algorithms deal with low-level aspects to be efficient. Our experience debugging memory corruptions in the Pharo VM shows that although VM crashes are sometimes reproducible in production scenarios, most of the time such reproductions are not deterministic and long to execute, crashes happen long after the corruption has appeared, and reproduction scenarios are large enough to be difficult to understand. Finding, reproducing, and debugging such bugs is complex and time-consuming (*c.f.* Section 2).

The Chromium Project finds that since 2015 70% of high-severity bugs are due to memory unsafety, and most of them are use-after-free errors (ChromiumProject). Existing work proposes to perform static verifications of GC algorithms using theorem provers (Sandberg Ericsson et al., 2019; Gammie et al., 2015) and model checking (Ugawa et al., 2017; Xu et al., 2022). Such approaches are heavy-weight to implement and execute, even at the expense of requiring specific techniques to optimize them (Abe et al., 2016) (*c.f.* Section 5).

In this article, we propose a lightweight test generation approach that automatically generates large sequences of random heap events guided toward a specific objective: testing ephemeron finalisation. Randomly generated events exercise the garbage collection algorithm with the objective of generating VM crashes and finding bugs. Once a bug is found, we use a test case reduction algorithm that finds the smaller subset of events that reproduces the issue (*c.f.* Section 3).

We implemented our approach on top of the VM simulator of the Pharo VM and used it to test the ephemeron finalization mechanism (Hayes, 1997) in Pharo's sequential stop-the-world generational scavenger (Ungar,

1984). We show that with our lightweight test generation approach 37% of generated cases produce failures in a matter of minutes. After automatic test case reduction and some manual inspection, we found that those failing configurations were provoked by two real-world bugs in Pharo's ephemeron implementation reproducible by trivial sequences of events that were easy to debug (*c.f.* Section 4).

**Paper contributions.** The contributions of this paper are:

- The first implementation, to the best of our knowledge, of a heap fuzzer that tests memory managers and garbage collection algorithms;

- Empirical evidence that a lightweight heap fuzzer implementation using simple random guided GC events is capable of finding real-world bugs, in this case in the Pharo Virtual Machine;

- An analysis showing that hitting failures with such an approach happens with high probability: generating 100 events reproduces such bugs more than 35% of the times, generating 50 events reproduces them around 27% of the times;

- Empirical evidence showing that such bugs can be reduced to trivial sequences that are easy to debug and synthesize as normal tests.

## 2 Garbage Collection Bugs

High-level object-oriented programming languages use automatic memory managers with garbage collection (GC) algorithms that are in charge of reclaiming memory (Jones, 1996). GC algorithms reclaim unused memory by observing the allocated memory chunks, and *proving* at runtime which ones are unreachable by the application graph. Depending on the underlying allocation mechanism, unreachable objects are *e.g.,* freed either by returning memory to the operating system or marking their memory as reusable.

Defects in garbage collection algorithms produce subtle effects that are revealed later in program execution as memory corruptions. The temporal distance between the cause and the failure makes these issues difficult to debug and understand. These problems are exacerbated by memory model extensions such as ephemeron finalization and weak references, and by the fact that garbage collection algorithms deal with low-level aspects to be efficient. Finding, reproducing, and debugging such bugs is complex and time-consuming.

**Use After Free Bugs.** Use-after-free bugs (CommonWeaknessEnumeration, c) are one of the most prominent bugs in today's applications. For example, they appear as one of the top 25 vulnerabilities in the Common Weaknesses Enumeration (CommonWeaknessEnumeration, b). As their name says, use-after-free bugs are produced when memory is used after being returned to the allocated. Although automatic memory managers reduce the risk of such errors by not letting developers explicitly free objects, such defects can still be present in a garbage collection implementation.

**Expired Pointer Dereference in Moving Collectors.** Expired pointer dereference bugs (CommonWeaknessEnumeration, a) happen when the object referenced by a pointer is moved or invalidated, typically when an object is moved but not all of its incoming references are updated. This kind of defect appears in moving collectors such as semi-space collectors, or compacting collectors. Moving collectors relocate objects in memory with the purpose of, for example, de-fragmenting/compacting the memory, improving locality, and putting together objects that are treated similarly (pinned objects, large objects, young objects...). After an object is relocated, all of its incoming references must be updated to the new location.

**Weak References and Ephemeron Finalization.** Weak references are references that do not prevent a garbage collector from collecting the referred object. If an object is only referenced by weak references it will be a candidate for reclamation. Related to weak references are ephemerons (Hayes, 1997), which are key-value pairs where the key is held strongly (in contrast to weakly) and the value is not traced unless the key is held strongly by some other object. Ephemerons are used as a finalization mechanism to notify the runtime when their key is a candidate for collection because they are the only object referencing it.

Ephemerons and weak references introduce many complexities in garbage collection algorithms. They need to be partially traversed or traversed separately from normal objects. Weak references may need to be replaced by tombstones marking the collection of an object. Object references in both ephemerons and weak references need to be properly updated in moving collectors. All these subtleties make testing such implementations even more challenging.

## 3    Fuzzing the PharoVM Finalization Mechanism

We propose to automatically test garbage collection algorithms by fuzzing heaps. Figure 1 illustrates our solution. The key idea is to generate large sequences of random events affecting the heap, with the objective of generating VM crashes and finding bugs. Event sequences are generated randomly by a fuzzer that guides generation toward a specific objective. Generated events are then executed on top of a VM with assertions enabled. The assertions present in the VM validate functions pre/post conditions and play the role of test oracle: we consider an event sequence as failing if it triggers an error or assertion failure.

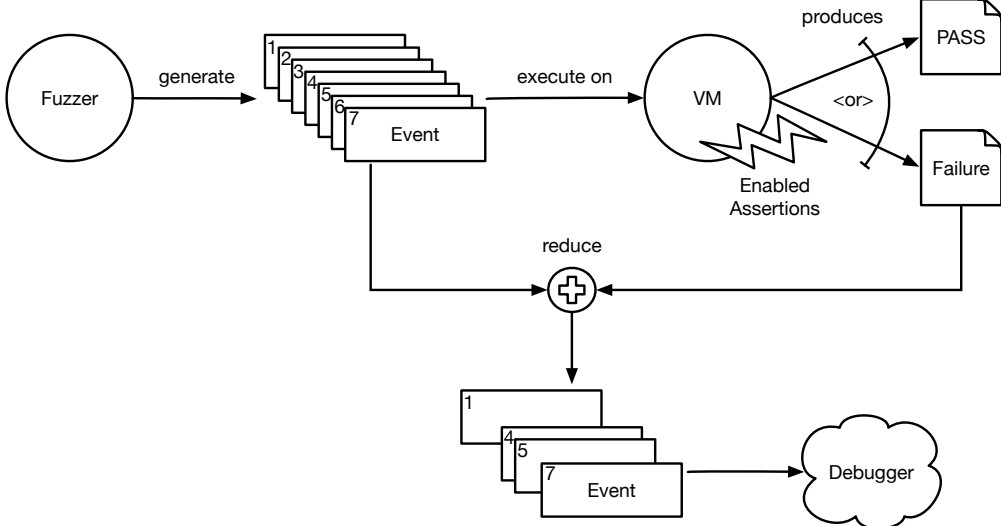

Figure 1: Solution overview. A fuzzer generates randomly guided events that are executed on a VM with assertions enabled. The execution either passes or produces a failure. Once a failure is discovered, the sequence of events is reduced and used to debug.

The event sequences are fixed once generated, making the observed issue reproducible by re-executing a sequence of events. Once a bug is found, we use a test case reduction algorithm that finds a smaller subset of events that reproduces the issue. Reduction helps in understanding the issue and finding its cause. It eliminates events that do not participate in the issue and that only add noise. Moreover, it produces small object graphs that are easy to follow and reason about.

### 3.1    Fuzzing Model

Events are generated as a stream. Each new event is initialized with random values and fixed afterward to guarantee reproducibility. The generation of each event has access to previously generated events. This allows mutation events to be configured with valid previously created allocations.

Our fuzzing model supports three different kinds of events as described below: object allocations, mutations, and garbage-collection runs.

**Allocation.** The allocation of an object. It is parameterized by the number of instance variables of the object, whether the object should be allocated in the new or the old space, and whether the object

must be a root. In our current implementation, all allocations create ephemeron objects. To make the memory model consistent, allocated objects need to have a class. Classes are created on a by-need basis, and all objects that have the same shape/format share the same class.

**Mutation.** The mutation of an object's instance variable. It makes an object reference another object (or itself). It is parameterized by a referrer and referee objects, and the assigned instance variable index. When a mutation event is created, it randomly takes one allocation event from current fuzzing to use as a referrer and one to use as a referee. The instance variable index is configured by creating a number at random between 1 and the number of instance variables of the referrer.

**Collection.** The execution of a garbage collection. It is parameterized with garbage collection arguments. In our current implementation, the only garbage collection argument is the level of the GC between one of the two following: (a) an incremental GC in the new space (a.k.a., a scavenge) and (b) a full GC implying a scavenge followed by a mark-and-compact of the old space with a single pass of compaction.

### 3.2 Assertions as Test Oracles

One of the biggest challenges in test generation is the generation of test oracles (Barr et al., 2015). A test oracle is a heuristic approach that decides if a given test execution finished with or without problems. Informally speaking, the oracle of a standard test case is the set of test assertions written by developers as in the example code below.

```
SetTestCase >> testNewSetIsEmpty
  | aSet |
  aSet := Set new.
  self assert: aSet isEmpty  "<− test oracle"
```

The complexity of test oracles for automatically generated test cases lies in the generation of meaningful assertions that capture the expected semantics of the test. In our case, we expect that generated tests (in the form of sequences of events) do not fail at runtime with exceptions, nor do they produce memory corruptions.

Our current model uses two heuristic oracles:

**VM Assertions.** The assertions inside the VM code check invariants such as pre and post-conditions. We rely on such autovalidating assertions to check the health of the GC algorithm. Figure 2 illustrates three assertions validating the full GC preconditions: that the mark stack and weakling stacks used for marking are valid and empty before the GC.

```
SpurMemoryManager >> globalGarbageCollect
  self assert: self validObjStacks.
  self assert: (self isEmptyObjStack: markStack).
  self assert: (self isEmptyObjStack: weaklingStack).
  ...
```

Figure 2: Orchestration of Test Execution. First, we initialize the heap. Then we execute all events on the current heap. Finally, we reset all events.

**Heap Invariant Checks.** The Pharo VM includes an invariant checker also called the leak checker. The invariant checker iterates all heap entities (objects, free memory chunks) and validates a series of invariants for each entity. For example, it validates that each object's class is not `nil`, that each of its slots references a valid object, and, if the object is old and refers to young objects, it validates that it is stored in the remembered set. We enabled the invariant checks to run on each GC.

### 3.3 Test Execution

Test execution is orchestrated by a Fuzzing object that contains all events for a particular fuzzed test. The execution of the test relies on three main steps, as illustrated in Figure 3. First, the fuzzing initializes a heap object, containing an underlying VM simulation. Second, we execute each event on the previously initialized heap. Such execution is stateful and produces side effects such as allocating objects and moving them. For simplicity, our implementation puts into events transient data such as the oop of allocated objects, to be remapped if moved (see Section 3.4). Finally, we reset all events to make the fuzzing reusable.

```
HFFuzzing >> basicExecute
  self prepareHeap.
  self events do: [ :event | event executeOn: self ]
  self events do: #reset.
```

Figure 3: Orchestration of Test Execution. First, we initialize the heap. Then we execute all events on the current heap. Finally, we reset all events.

### 3.4 Remapping Objects and Invalid Events

Given the model above, mutation events can become invalid in two cases. First, at construction time, if a mutation is created before any allocation event happens, no mutation is possible. The event builder has then the possibility to react by either creating one allocation or producing a *nop* event without effect. Our current prototype implements the latter.

Second, when executing the stream of events a mutation may have been statically configured to perform an assignment from/to an object that was collected by a previous collection event. This happens because the event builder cannot statically predict if an allocation remains valid during construction. Predicting object collections at build time would require duplicating the GC semantics in the builder.

Instead, our solution detects collected objects and remaps their oops after each GC. For each allocation, we keep track of its oop[1] and its identity hash. After a GC, we iterate the heap and build a relocation map ($identityhash => oop$) that we use to remap the oops above and mark collected objects. This approach showed practical because the generated heaps contain in the order of tens of objects.

### 3.5 Test Case Reduction

Once a bug is found we use the delta debugging algorithm to reduce the event sequence. The objective of this reduction is to simplify further debugging and speed up the test execution. We implemented in our prototype the *ddmin* delta debugging algorithm (Zeller & Hildebrandt, 2002). This algorithm reduces the input successively by removing events while the issue is reproduced. When one removed event makes the issue disappear, the algorithm restores the removed event, backtracks, and continues with the removal of other events.

### 3.6 Limitations

Although this paper performs random testing of a GC algorithm, the current solution does not necessarily achieve a high code coverage of the search space. Our solution guides fuzzing towards a part of the GC and memory manager. This, of course, does not preclude the existence of other bugs.

---

[1] ordinary object pointer

## 4   Evaluation

We implemented our approach on top of the VM simulator of the Pharo VM (Miranda et al., 2018) and using the pre-existing heap initialization of the Pharo VM testing infrastructure (Polito et al., 2021). The implementation is hosted in https://github.com/Alamvic/heapFuzzer and is licensed MIT.

Our main objective was to test the ephemeron finalization mechanism in the Pharo VM GC. The Pharo VM GC is a sequential stop-the-world generational scavenger using a mark-compact algorithm for older generations. We show how our prototype found two major bugs in Pharo's ephemeron implementation and one bug in the compaction algorithm, and we show how such test cases were automatically reduced to trivial sequences that were easy to debug.

We run our approach 100 times on the following configuration:

- Pharo Version: Pharo 11 alpha build #219, commit ccb9b558742bf1d86df35c44ae5802e4a05a03ca

- Pharo VM commit: 3172ed61ee1accbccdd11130e0e797a106ca4e69

- HeapFuzzer commit: f37c235a91205ad51630a918a9943d727991d8df

We chose 100 as timeout because in the experiments below it turned out to be a reasonable timeout. First, it makes the solution practical by running fast. Second, we found most of the bugs that were of interest to us.

### 4.1   Statistics

**Implementation complexity.**   We implemented our solution in just above 345 lines of code, measured in the following way:

```
('HeapFuzzer' asPackage classesForClassTag: 'Core') sum: [ :e | e linesOfCode ]
>> "345"
```

**Probability to hit a bug.**   We measured the probability of hitting a bug in our setting, by running our approach 100 times. Our approach detected a bug 37 times out of 100.

**# events to hit a bug.**   We measured how many events we had to execute before hitting a bug for each of the 37 failing events. Figure 4 shows a histogram with 10-sized buckets representing the number of failing runs on events 1-10, 11-20, 21-30, and so on. The histogram shows that occurrences are mostly concentrated at the beginning of the runs, with ∼72% of chances of hitting a bug within 50 events in those that fail.

**Average test size after reduction.**   After reducing all 37 failing events, all bugs were reduced to sequences of 4-5 events, as shown in Figure 5. As shown in the figure, 27 cases failed on 4 events. The global average is 4.27 events per failure.

**Bug symptoms and causes.**   We manually analyzed the 37 failing executions and found that the failures happened in 5 different code locations *i.e.,* the 37 executions revealed 5 different errors/assertion failures. After test reduction, error locations remained the same. Those 5 failures were produced by 3 unique defects, two severe bugs producing memory corruptions described in Section 4.2 and Section 4.3 and a non-severe broken invariant described in Section 4.4.

### 4.2   Severe Bug 1: Marking Ephemerons

Our first discovered bug, presented only four events after reduction: two *root* allocations in the new space, a mutation, and a full GC. The test discovered with the invariant checker that the second ephemeron's class was mistakenly garbage collected producing a memory corruption. Figure 6 shows the test case reproducing the bug as we transcribed it from our fuzzing model to a normal test case.

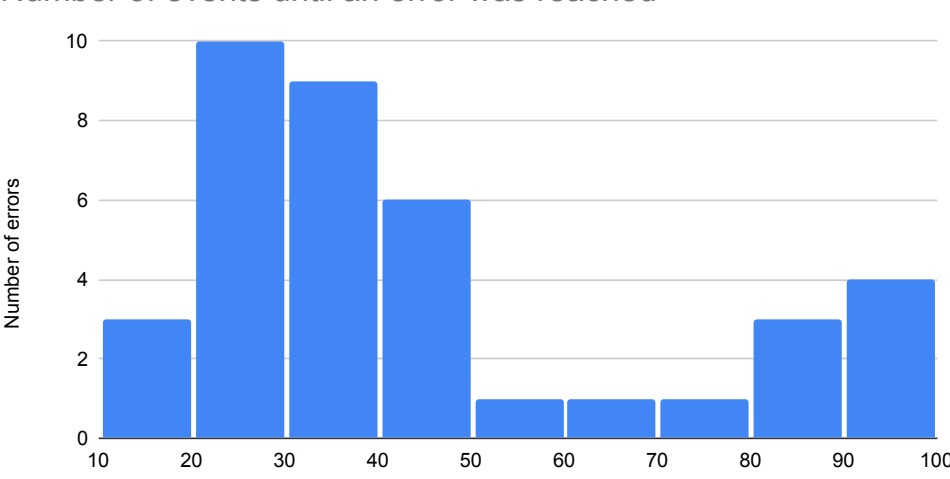

Figure 4: Number of events that needed to be executed before hitting a bug.

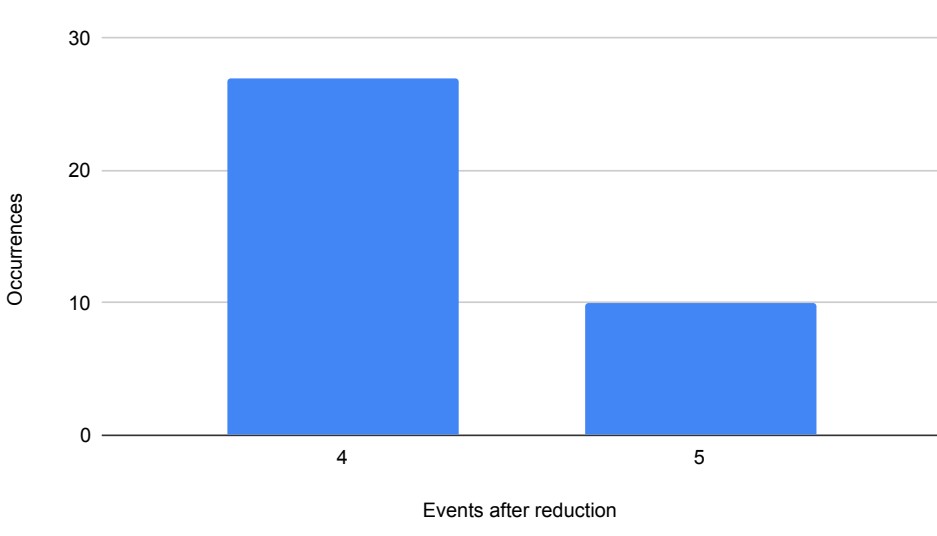

Figure 5: Number of events that needed to be executed before hitting a bug after reduction.

### 4.3   Severe Bug 2: Scanning Ephemeron References

Our second discovered bug, presented also four events after reduction: one *root* allocation in the new space, one non-root allocation in the new space, a mutation, and a full GC. The test discovered with an assertion during pointer update that the GC was not scanning ephemeron value references, producing a memory corruption. Figure 7 shows the test case reproducing the bug as we transcribed it from our fuzzing model to a normal test case.

```
testNewRootEphemeronHoldsAnotherEphemeronAsKeyThenFullGC
  | ephemeron1 ephemeron2 |
  ephemeron1 := self newEphemeronObjectWithSlots: 5.
  self keepObjectInVMVariable1: ephemeron1.
  ephemeron2 := self newEphemeronObjectWithSlots: 3.
  self keepObjectInVMVariable2: ephemeron2.
  memory storePointer: 0 ofObject: ephemeron1 withValue: ephemeron2.

  memory setCheckForLeaks: 63 "all".
  memory fullGC
```

Figure 6: Reduced test for bug #1. Two root ephemerons in the new space. One has the other as key. The GC was not marking/scanning the second ephemeron leading to the collection of it's class.

```
testNewRootEphemeronHoldsOldNonRootEphemeronAsNonKeyThenFullGC
  | ephemeron1 ephemeron2 |
  ephemeron1 := self newEphemeronObjectWithSlots: 17.
  self keepObjectInVMVariable1: ephemeron1.
  ephemeron2 := self newOldEphemeronObjectWithSlots: 7.
  memory storePointer: 1 "Not the key" ofObject: ephemeron1 withValue: ephemeron2.

  memory setCheckForLeaks: 63 "all".
  memory fullGC
```

Figure 7: Reduced test for bug #2. One root ephemeron in the new space and one non-root in the old space. The first one has the other as a value. The GC was not updating the references of the first ephemeron to the second ephemeron.

### 4.4 Bug 3: Compacting Unmarked Object at End of memory

Our third discovered bug, presented five events after reduction: one *root* allocation in the new space, a full GC, one non-root allocation in the new space, a mutation, and another full GC. The test discovered with the invariant checker that an unmarked object remained in the old space after compaction, *i.e.,* the object was not compacted and returned to free memory. Figure 8 shows the test case reproducing the bug as we transcribed it from our fuzzing model to a normal test case.

```
testNewRootEphemeronIsHeldsByOldNonRootEphemeronAsNonKeyThenFullGC
  | ephemeron1 ephemeron2 |
  ephemeron1 := self newEphemeronObjectWithSlots: 6.
  self keepObjectInVMVariable1: ephemeron1.

  memory fullGC.
  ephemeron1 := self keptObjectInVMVariable1.

  ephemeron2 := self newOldEphemeronObjectWithSlots: 15.
  memory storePointer: 10 "Not the key" ofObject: ephemeron2 withValue: ephemeron1.

  memory setCheckForLeaks: 63 "all".
  memory fullGC
```

Figure 8: Reduced test for bug #3. One root ephemeron in the new space and one non-root in the old space. The second one has the first as a value. The GC was not compacting unmarked objects at the end of the memory.

## 5   Related Work

### 5.1   Automated Garbage Collection Testing and Verification

The testing and verification of garbage collection algorithms have been mostly approached with formal verification. Existing work is divided into two families. One family of solutions performs static verifications of GC algorithms using theorem provers (Sandberg Ericsson et al., 2019; Gammie et al., 2015). A second family of solutions implements model checking approaches (Ugawa et al., 2017; Xu et al., 2022).

The first difference with our approach is complexity. Formal verification is heavy-weight to implement and execute. It requires the specification of GC semantics as a separate formalism that requires maintenance to stay in sync with the actual implementation. Thus, techniques have been developed to optimize formal approaches (Abe et al., 2016). In comparison, our solution is lightweight to implement, replicate and execute.

Another difference between existing approaches and ours is that they mostly focus on the verification of concurrent GCs. In this work, we focused on the verification of a sequential stop-the-world GC, and specifically on its ephemeron implementation. While in the future we plan to use our approach to test concurrent GCs, our approach is applicable to partial features of a GC implementation without the need for an entire memory model.

### 5.2   Automated Virtual Machine Testing

Besides work on validating GCs, a lot of work focuses on testing VMs in general, and more specifically JIT compilers. The teams of Maxine and Pharo reported recently QEMU-based unit testing infrastructures for cross-ISA testing and debugging (Kotselidis et al., 2017; Polito et al., 2021; Ducasse et al., 2022). They reported that these infrastructures helped them in porting their VMs to ARMv7 and ARMv8 64bits respectively. Although their approaches are based on unit testing, they rely on manually written tests, while our approach performs automatic test generation.

Lately, several works have explored the path of program fuzzing and differential testing. Recent work explores the generation of test cases for JIT compilers using available interpreters (Polito et al., 2022). Other solutions make use of differential testing between different Virtual Machines for a single language. Several works explore bytecode fuzzing on the Java Virtual Machine (JVM) (Chen et al., 2016; 2019) or test generation for JVM native extensions (*i.e.,* JNI) (Hwang et al., 2021). Similar work appeared recently for JavaScript engines using test transplantation (Lima et al., 2020) and compiler fuzzing (Ye et al., 2021; Park et al., 2021). Although we share with these approaches the goal of automatic test generation, these explore JIT compiler testing and treat VMs as black boxes. Instead, our fuzzing approach is gray-box: it has knowledge of the GC behavior to generate pertinent events. Moreover, our approach produces random yet reproducible tests that exercise the GC, while the other approaches are subject to VM non-determinisms in the JIT compilers and their profiling mechanisms.

### 5.3   VM Simulation Environments and Meta-circular VMs.

Meta-circular VMs and VM frameworks have offered simulation environments for a long time. Such simulation environments help in testing and debugging virtual machines. Such is the case of Self (Ungar et al., 2005), Smalltalk (Ingalls et al., 1997; Miranda et al., 2018) and Maxine (Wimmer et al., 2013). Our solution exploits simulation environments to simplify the execution of generated unit tests.

## 6   Conclusion

In this article, we proposed to fuzz heaps by generating large sequences of randomly guided heap events. Such events exercise the GC algorithm to produce VM crashes and find bugs. Once a bug is found, we use a test case reduction algorithm that finds a small subset of events that reproduces the issue.

We implemented our approach on top of the VM simulator of the Pharo VM and showed that our prototype is able to find real-world bugs quickly. We further described three bugs our solution found in Pharo's garbage

collection: two severe ephemeron implementation bugs that produced memory corruptions, and one bug in the compaction algorithm that prevented the compaction of some objects at the end of the heap. In the future, we plan to extend the approach to manage other than ephemerons, and to use it to evaluate the correctness of other GC algorithms.

**Acknowledgments**

This work was funded by Inria's Action Exploratoire AlaMVic.

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
