# OpenReview forum: "Testing Finalization Mechanisms with Heap Fuzzing"
_FAST.org.ar/2022/Workshop — FAST Smalltalk 2022_

### Official Review · Reviewer_8pwR · 2022-10-21
**Interesting acount on how 345 lines of code uncovered 3 hidden & long standing GC bugs involving ephemerons**

**Rating:** 7
**Confidence:** 5

**Review:**

The paper describes how randomly generated events were used to explore the space of object memory configurations pursuing the violation of invariants and deducting from them unit tests that exposed bugs related to the reclamation of ephemerons. The strategy suggests that the same ideas could be applied to the debugging of other aspects of the GC.

—

Generally speaking, the use of uniformly generated random data is not sufficient for hitting the thing one is looking for: some other technique is needed to incrementally conduct the sampling. In this case, however, this simple approach, which showed to be effective, is also welcome because any influence on the sampling could have favored the reproduction of one kind of sequencing over others, presumably less frequent in practice but harder to find for this very reason.

—

In pg. 3 it reads: "*The event sequences are fixed once generated, making the observed issue reproducible*". Usually, sound sampling techniques are able to repeat any sample by just re-injecting the saved seed to the random generator. Why is the entire sequence of events preserved instead?

If "*all allocations create ephemeron objects*", then all referrers and referees used by mutation events are ephemerons. What's the rationale behind this restriction? Even in this case, where we are focused on verifying the collection of ephemerons, why should we expect that this setting provides a good coverage of cases? e.g., strong -> ephm, weak -> ephm, ephm -> ephm etc.

VM Assertions: it would be good if the paper provided some examples of these assertions (much as it mentions some Heap Invariant Checks).

Figure 3 could indicate that the distribution of errors is bimodal, in fact on the right end, the number of errors starts to increase for a second time. Why was the number of events limited to 100? Couldn't this limit be hiding some more errors occurring to the right of the 100th event?

In 4.2 it reads: "*the second ephemeron’s class was mistakenly garbage collected*". I must presume that these classes were not installed in any global root. So, why did the error manifest with the class and not with other slots? Is it just because the instance variables had non-oop objects such as SmallIntegers or something of that sort? I would also like to know whether both defects were independent or somehow related.

Did the selection of a random number of instance variables play any role? I would have considered such a variation to be crucial in a more general event setting aimed at testing other parts of the GC. Here, however, I tend to see this variation as *noise*, i.e., not being actually representative of new territories in the configuration space. Wouldn't it have been equally effective if objects had 1 atRandom slots (with some possible addition such as large objects of a fixed size)?

—

Typo in Bug 2, page 7: the text should refer to Figure 6 and not 5.

---

### Official Review · Reviewer_Yqbb · 2022-11-01
**Testing Finalization Mechanisms with Heap Fuzzing**

**Rating:** 9
**Confidence:** 5

**Review:**

Very well written. Only a few grammar errors:
"the smaller subset" -> "a smaller subset" or "the smallest subset"
"collector to collect" -> "collector from collecting"
"generated heaps present objects in the order of the tens" -> "generated heaps contain on the order of tens of objects"

Figure 3, the ".00" on the x-axis labels are at best irrelevant, and at worst confusing.

In sec 4.3 and 4.4 "root" is italic, but not in 4.2 - for unknown reason.

"offer simulation environments" -> "have offered simulation environments"

This is an excellent paper that outlines an important problem, proposes a novel solution, and demonstrates the efficacy of the solution. In particular, the pruning of sequences is very important and remarkably successful, at least in the cases described.